# Synthetic CRISPR-Cas gene activators for transcriptional reprogramming in bacteria

Chen Dong[1], Jason Fontana[2], Anika Patel[1], James M. Carothers [2,3,4] & Jesse G. Zalatan [1,2,4]

Methods to regulate gene expression programs in bacterial cells are limited by the absence of effective gene activators. To address this challenge, we have developed synthetic bacterial transcriptional activators in *E. coli* by linking activation domains to programmable CRISPR-Cas DNA binding domains. Effective gene activation requires target sites situated in a narrow region just upstream of the transcription start site, in sharp contrast to the relatively flexible target site requirements for gene activation in eukaryotic cells. Together with existing tools for CRISPRi gene repression, these bacterial activators enable programmable control over multiple genes with simultaneous activation and repression. Further, the entire gene expression program can be switched on by inducing expression of the CRISPR-Cas system. This work will provide a foundation for engineering synthetic bacterial cellular devices with applications including diagnostics, therapeutics, and industrial biosynthesis.

[1] Department of Chemistry, University of Washington, Seattle, WA 98195, USA. [2] Molecular Engineering & Sciences Institute, University of Washington, Seattle, WA 98195, USA. [3] Department of Chemical Engineering, University of Washington, Seattle, WA 98195, USA. [4] Center for Synthetic Biology, University of Washington, Seattle, WA 98195, USA. Correspondence and requests for materials should be addressed to J.M.C. (email: jcaroth@uw.edu) or to J.G.Z. (email: zalatan@uw.edu)

Bacteria are attractive targets for a wide variety of engineering applications. Bacterial strains with the ability to utilize carbon sources like $CO_2$, CO, methane, or lignocellulose, and alternative energy sources such as light or $H_2$ could provide the foundation for cost-effective and environmentally-friendly industrial biosynthesis[1,2]. Microbial communities, such as those that reside in the human gut, play an important role in human health and disease, and tools to engineer these bacteria have great potential as both diagnostics and therapeutics[3–5]. To harness, regulate, and modify the behavior of these and other bacteria, there is a compelling need to develop genetic tools to control gene expression and implement complex, multi-gene regulatory programs. Ideally, we want to build circuits that can regulate many genes at once, dynamically respond to external inputs or the internal state of the cell, and be easily reprogrammed to explore different functional architectures. While capabilities to edit and modify genomes are rapidly expanding, our ability to encode a precisely-defined and dynamically-responsive gene expression program with cis-regulatory sequences at the DNA level remains difficult. Thus, we sought to develop synthetic transcription factors in bacteria, which could be coupled to programmable DNA binding domains and controlled by inducible promoters to engineer complex, dynamically-responsive multi-gene expression programs.

Synthetic control of gene expression has recently become much more straightforward with the emergence of programmable transcription factors using the CRISPR-Cas system (Fig. 1). A catalytically-inactive Cas9 (dCas9) protein can be used to target specific DNA sequences with guide RNAs (gRNAs) that recognize their targets based on predictable Watson-Crick base pairing. This approach can be used to repress genes by physically blocking RNA polymerase (CRISPR interference or CRISPRi)[6,7]. To activate genes (CRISPR activation or CRISPRa), the CRISPR complex can be linked to a transcriptional activator by direct fusion to dCas9 or via recruitment domains on the gRNA[7–11]. In bacteria, however, there are very few transcriptional activation domains that have been reported to be effective when fused to modular DNA binding domains. Bacterial two-hybrid systems have been constructed with pairs of candidate interacting proteins separately fused to RNA polymerase subunits and DNA binding proteins. It is also possible to fuse RNA polymerase subunits directly to DNA binding domains to activate transcription[12–14]. One of the RNA polymerase subunits, RpoZ, has been coupled to the CRISPR system to activate gene expression[7,15–18]. For comparison, in eukaryotic systems there are many effective activators and CRISPRa has been extensively used in a variety of applications[19]. The paucity of reports of CRISPRa

in bacteria suggests that RpoZ may not be effective as a general activator of transcription, or that we lack a complete understanding of the design rules to predictably activate gene expression in bacteria.

To develop an improved toolkit for gene activation in bacteria, we screened a broad set of candidate proteins for transcriptional activity in E. coli. We identified several proteins that can effectively activate gene expression when recruited via the CRISPR-Cas system. Using the most effective activator, SoxS, we can activate one target gene while simultaneously repressing a different target gene with CRISPRi, and we can control the entire multi-gene expression program with inducible promoters driving CRISPR-Cas system components. We find that gene activation in E. coli is highly sensitive to the location of the gRNA target site, consistent with prior results[7], and suggesting a possible explanation for why it has been difficult to develop synthetic activators in bacteria. Finally, we show that bacterial CRISPRa can be used to increase the output of a heterologous ethanol biosynthesis pathway. These results provide a framework for implementing CRISPRa in bacteria with a wide variety of potential applications. Further, because SoxS interacts with a highly conserved site on RNA polymerase, our bacterial CRISPRa toolkit may be portable to a broad range of bacterial species.

## Results

**Identifying transcriptional activation domains for bacteria.** To recruit transcriptional activators to the CRISPR-Cas system, we used gRNAs that are extended with hairpin sequences to recruit RNA binding proteins (RBPs), which are in turn fused to candidate activators (Fig. 1a)[10]. These modified gRNAs, termed scaffold RNAs (scRNAs), encode both the target sequence and the regulatory action to execute at that target. By using scRNAs to recruit activators (CRISPRa) to some genes and gRNAs to physically block RNA polymerase (CRISPRi) at other genes, we can encode complex expression programs where some genes are activated and others are repressed (Fig. 1b)[10].

To screen for potential activators, we first constructed an E. coli strain with a genomically-integrated, weakly-expressed GFP reporter gene. The upstream region of the reporter gene includes a number of potential gRNA target sites and is identical to that used previously to evaluate the dCas9-RpoZ fusion protein (see Supplementary Methods)[7]. We targeted the CRISPR-Cas complex to the W108 gRNA site located 91 bases upstream of the transcriptional start site (TSS), as this site previously demonstrated the strongest activation with dCas9-RpoZ[7]. For candidate activators we chose endogenous transcriptional regulators SoxS,

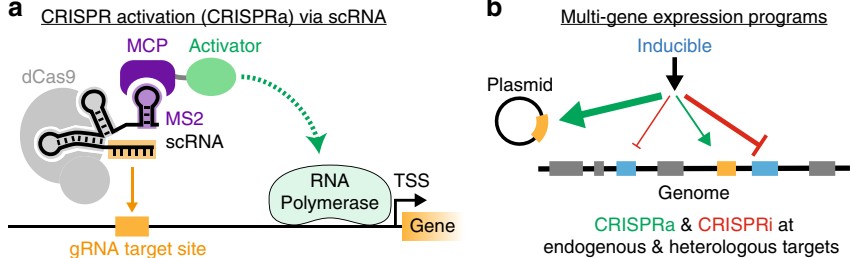

**Fig. 1** CRISPR activation in bacteria enables complex multi-gene expression programs. **a** To activate gene expression, we target a CRISPR-Cas complex upstream of a target gene. dCas9 binds a scaffold RNA (scRNA), which is a modified gRNA that encodes both the target sequence and an RNA hairpin to recruit effector proteins that interact with RNA polymerase. The schematic depicts a 1x MS2 scRNA containing an MS2 RNA hairpin, which binds the MS2 coat protein (MCP) that is fused to candidate activator proteins[10]. **b** Combining CRISPRi with CRISPRa enables multi-gene expression programs for simultaneous activation and repression. scRNAs that recruit activators can target genes for activation, while gRNAs targeted within a gene result in CRISPRi-based repression. If the CRISPR-Cas system components are controlled by inducible promoters, the entire gene expression program can be dynamically regulated

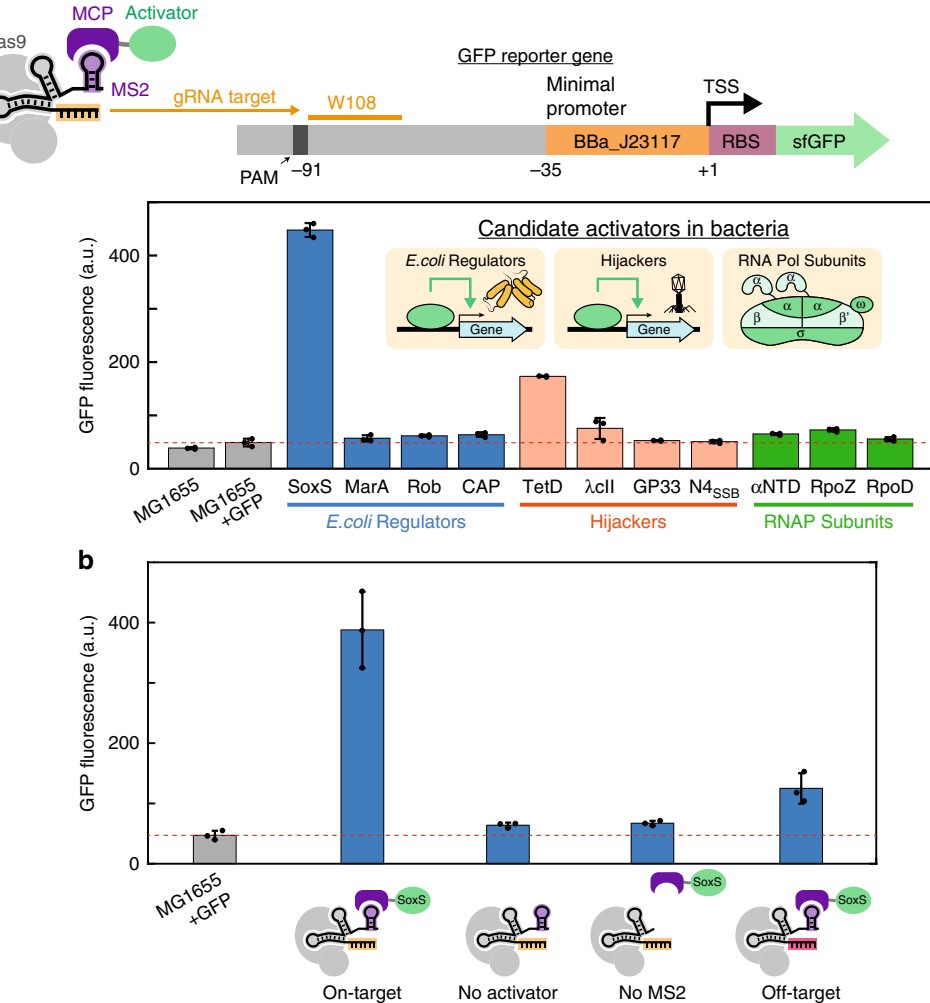

**Fig. 2** Effector proteins can activate reporter gene expression in *E. coli*. **a** The CRISPRa complex targets a GFP reporter gene (sfGFP, superfolder GFP) driven by a weak BBa_J23117 promoter. The gRNA target is the W108 sequence, located 91 bases upstream of the transcription start site (TSS). Several candidate activator proteins fused to MCP result in significant increases in GFP expression, including SoxS and TetD. The dotted red line indicates the background fluorescence level observed in the parental MG1655 *E. coli* strain containing the GFP reporter gene. GFP levels were measured in late stationary phase. Similar trends with smaller overall effects were observed in exponential phase. **b** All components of the CRISPRa complex must be present for gene activation. Significant GFP expression is observed when dCas9, the 1x MS2 scRNA, and MCP-SoxS are all expressed. When MCP-SoxS is omitted (no activator) or the MS2 hairpin is removed (no MS2), there is no significant GFP expression. Modest GFP expression is observed when an off-target scRNA is used that has no target site in this strain (RR2, Supplementary Table 2), suggesting that overexpression of SoxS may have some off-target gene activation effects. Values reported are GFP fluorescence levels measured by flow cytometry. Values are median±s.d. for at least three biological replicates (specific values are indicated by black dots)

MarA, Rob, and CAP[20,21]; hijackers (i.e. bacteriophage or transposon effectors) TetD, λcII, GP33, N4_SSB, and AsiA[22–26]; and RNA polymerase subunits RpoZ (ω), RpoD (σ[70]), and the N-terminal domain of RpoA (αNTD)[12,13,27]. Of these candidates, αNTD[12], RpoZ[7,13], and AsiA[26] have been previously reported to activate transcription when fused to heterologous DNA binding domains, and other candidates were selected based on literature reports suggesting that they could recruit RNA polymerase to activate transcription. Each of these candidate proteins was fused to the MS2 coat protein (MCP), which binds to an MS2 hairpin on the scRNA (Fig. 2a)[10].

Several candidate activators produced significant GFP reporter expression in late stationary phase cultures, with the largest effects from SoxS and TetD, and smaller but still detectable GFP expression from λcII, αNTD, and RpoZ (Fig. 2a). Activation with MCP-RpoZ is significantly increased in a Δ*rpoZ* host strain, consistent with that observed previously for other RpoZ fusion proteins, including dCas9-RpoZ (Supplementary Fig. 1a)[7,13]. We also obtained significant GFP expression with the T4 bacteriophage activator AsiA (Supplementary Fig. 1b) by co-transforming with a σ[70] F563Y mutant, which prevents toxicity that occurs when AsiA is expressed alone and inhibits the endogenous σ[70] subunit[26]. The most effective activator, SoxS, is a member of the AraC family of transcription factors. SoxS is expressed during oxidative stress and activates expression of a number of genes by binding to RNA polymerase in a pre-recruitment complex and scanning the genome to find DNA targets located in promoter regions[20,28]. This mechanism of action could explain the effectiveness of SoxS as a candidate activator for CRISPRa.

To confirm that the observed GFP expression arises from recruitment of the candidate activator upstream of the GFP reporter gene, we performed several negative controls with the SoxS activator. First, we demonstrated that activation requires the

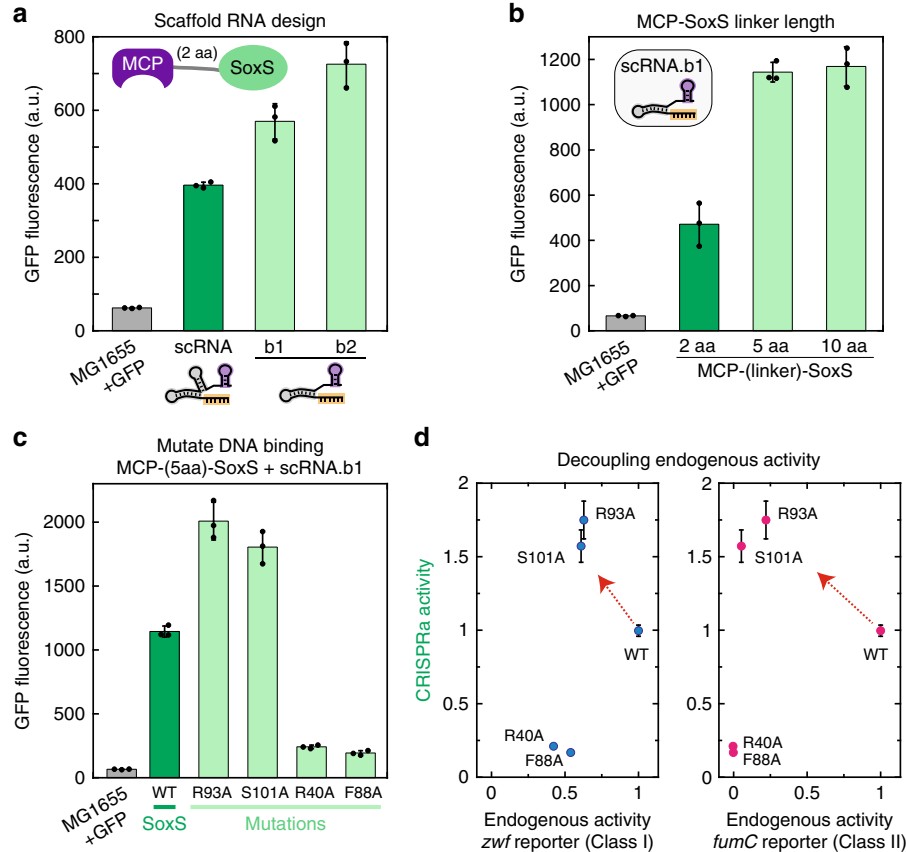

**Fig. 3** Optimization of gene activation. **a** Modifying the scRNA design to remove the tracr terminator hairpin results in a 1.5–2-fold increase in gene expression. Complete sequences of the original scRNA, scRNA.b1, and scRNA.b2 designs are included in the Supplementary Methods. scRNA.b1 and scRNA.b2 differ by one base from the 5′ end of the terminator hairpin. **b** Increasing the linker length between MCP and SoxS from 2 to 5 or 10 amino acids increases GFP expression by 2–3-fold. **c** Mutations in SoxS at the binding interface to endogenous DNA targets have variable effects on activity in a CRISPRa assay. Point mutations at SoxS R93A or S101A result in a 2-fold increase in GFP expression, while SoxS R40A and F88A lead to substantial decreases in GFP expression. Based on the structure of the SoxS homolog MarA[33], residues R93, S101, and R40 are surface exposed while the F88 side chain points into the hydrophobic core. **d** Plots of CRISPRa activity vs endogenous activity for wild type (wt) and mutant SoxS proteins indicate that transcriptional activation can be decoupled from binding to endogenous targets. CRISPRa activity values are GFP fluorescence levels (**c**) normalized to the value obtained for wt SoxS. Endogenous activity values are from LacZ reporter assays with endogenous SoxS promoters for the *zwf* and *fumC* genes (Supplementary Fig. 3a, b), corrected for background reporter activity in the absence of SoxS and normalized to the value obtained for wt SoxS. *zwf* is representative of class I SoxS target genes in which the SoxS site is located upstream of the −35 site, and *fumC* is representative of class II SoxS target genes in which the SoxS site overlaps the −35 site (Supplementary Fig. 3a). The data shown for CRISPRa and endogenous activity were obtained in separate reporter strains. Similar results were obtained when CRISPRa values were measured in strains with both the GFP and LacZ reporters (Supplementary Fig. 3c). Values reported are GFP fluorescence levels measured by flow cytometry. Values are median±s.d. for at least three biological replicates (specific values are indicated by black dots)

presence of the activator protein. When dCas9 and the 1x MS2 scRNA are expressed without MCP-SoxS, the CRISPR-Cas complex can bind upstream of the GFP reporter, but there is no significant GFP expression (Fig. 2b). Similarly, when the gRNA lacks the 1x MS2 recruitment hairpin, there is no significant GFP activation. Finally, we expressed dCas9, MCP-SoxS, and an off-target 1x MS2 scRNA and observed weak but detectable GFP expression (Fig. 2b). This result suggests that a small fraction of the GFP expression observed with MCP-SoxS arises from non-specific gene activation, which is plausible given that SoxS is an endogenous regulator of transcription in *E. coli* with many gene targets and a relatively degenerate binding site[29,30].

**Optimizing activity of the SoxS activator.** To optimize the activity of CRISPRa with SoxS and minimize off-target effects at endogenous SoxS gene targets, we systematically varied several design parameters of our system. First, we modified the scRNA

structure to optimize it for bacterial expression. In our original design, the MS2 hairpin is appended at the 3′ end of the gRNA sequence[10], just downstream of an endogenous tracr terminator hairpin[31]. Removing this terminator hairpin while retaining the 3′ MS2 hairpin leads to a 1.5–2-fold increase in GFP expression, likely due to increased steady-state levels of the full length scRNA (Fig. 3a). For comparison, in eukaryotic cells the same modified scRNA lacking the terminator reduces CRISPRa-mediated reporter gene expression ~2-fold (Supplementary Fig. 2a), possibly because the terminator hairpin interacts with Cas9[32]. Presumably, the same detrimental effect is present in bacteria, but it is outweighed by the benefit from ensuring that the entire scRNA construct is expressed. Using this optimized scRNA design (scRNA.b1), we then varied the length of the amino acid linker connecting MCP and SoxS. 5 and 10 amino acid linkers increased GFP expression by ~2–3-fold compared to our original 2 amino acid linker (Fig. 3b). To increase activity further, we tested an scRNA design with two MS2 hairpins (2x MS2), but observed no significant increase in GFP expression

compared to the 1x MS2 scRNA (Supplementary Fig. 2b). We also tested an alternative gRNA design in which MS2 hairpins are embedded within existing hairpins in the gRNA (sgRNA 2.0)[11]. We observed no significant GFP expression with sgRNA 2.0 (Supplementary Fig. 2c), which was surprising given that this design is highly effective in eukaryotic cells.

To minimize potential off-target effects at endogenous SoxS gene targets, we attempted to decouple its endogenous DNA binding activity from its transcriptional activation function. We identified several candidate residues at the SoxS DNA binding interface that are known to disrupt activity at endogenous sites when mutated[33,34]. We expected that if transcriptional activation could be decoupled from endogenous DNA binding, we would retain activity with our CRISPRa system even with SoxS mutants that are defective for activity at endogenous promoters. For two SoxS mutants, R93A and S101A, we observed no loss in activity in a CRISPRa assay; rather, we observed a 2-fold increase in GFP expression compared to wild type (wt) SoxS (Fig. 3c). The increased activity could arise if less SoxS is sequestered at endogenous DNA sites in these mutants. In contrast, the mutations R40A and F88A result in a substantial loss of GFP fluorescence in a CRISPRa assay, which could result from perturbations to protein structure or stability. To assess whether we effectively decoupled transcriptional activation from endogenous DNA binding, we plotted CRISPRa versus endogenous promoter activity for the series of SoxS mutants (Fig. 3d). To directly measure endogenous activity at SoxS gene targets when MCP-SoxS fusion proteins are expressed, we used LacZ fusion constructs for two different SoxS target genes, *zwf* and *fumC* (Supplementary Fig. 3), which are representative of class I and class II SoxS target genes, respectively[34]. The mutants that retain CRISPRa function, R93A and S101A, are ~2-fold weaker in endogenous activity than wt SoxS at *zwf* and >4-fold weaker at *fumC*. Thus, with SoxS R93A or S101A, we can partially decouple transcriptional activation from endogenous DNA binding. It may be possible to further decouple transcriptional activation from DNA binding with additional SoxS mutations.

We proceeded to test a fully optimized system with the 1xMS2 scRNA.b1 design, a longer 5 amino acid linker between MCP and SoxS, and the SoxS R93A mutant to reduce non-specific activity at endogenous SoxS targets. We observed a substantial increase in GFP fluorescence using an on-target guide RNA and a significant decrease in background activity with an off-target guide RNA (Supplementary Fig. 4) compared to the original, unoptimized system (Fig. 2b). Using RT-qPCR, we observed a 50-fold increase in GFP mRNA levels relative to negative control strains (Supplementary Fig. 5). Importantly, we observed no significant growth burdens associated with expression of the CRISPRa system (Supplementary Fig. 6a, b). Further, while we typically measured GFP levels in late stationary phase, we found similar trends in gene activation with smaller overall effects when GFP levels were measured in exponential phase (Supplementary Fig. 6c). We therefore proceeded with MCP-(5aa)-SoxS$_{R93A}$ and 1x MS2 scRNA.b1 in future experiments. In some cases, we also used the functionally equivalent 1x MS2 scRNA.b2 design (Fig. 3a), which differs by one base (see Supplementary Methods). In direct comparisons, this fully optimized SoxS-based CRISPRa system performs better than the previously reported dCas9-RpoZ bacterial CRISPRa system[7]. Activation with SoxS is effective in an MG1655 strain where dCas9-RpoZ activity is undetectable, and in a ΔrpoZ MG1655 strain the SoxS-based system outperforms dCas9-RpoZ by >2-fold (Supplementary Fig. 7).

**Inducible and simultaneous control of multiple genes**. With CRISPR-Cas transcriptional programs, it is straightforward

to target multiple genes for activation or repression[6,8,10,35], although simultaneous activation and repression has not yet been demonstrated in bacteria. Using our optimized bacterial CRISPRa system, we tested whether we could simultaneously activate one gene while repressing another. We constructed an *E. coli* strain with two genomically-integrated fluorescent reporter genes, a weakly expressed GFP and a strongly expressed RFP. We expressed dCas9, MCP-(5aa)-SoxS$_{R93A}$, a 1x MS2 scRNA.b1 for GFP activation, and a gRNA targeting RFP for CRISPRi repression (Fig. 4a). When both the scRNA and the gRNA are expressed, we observe a simultaneous increase in GFP expression and decrease in RFP expression, and the magnitudes of the effects are indistinguishable from those observed when each gRNA is expressed alone (Fig. 4a). Thus, when all components are expressed constitutively, we can effectively control multiple genes simultaneously and have different effects at individual gene targets.

To develop the capability to dynamically regulate multi-gene programs, we first attempted to control the CRISPRa system with inducible promoters. In previous work, using Tet-inducible promoters to control CRISPRi in bacteria required pTet controlling both dCas9 and the gRNA[6]. We tested whether a similar strategy could effectively control CRISPRa. When dCas9 and the 1x MS2 scRNA are controlled by pTet and MCP-(5aa)-SoxS$_{R93A}$ is expressed constitutively, however, we observed leaky GFP expression in the absence of the anhydrotetracycline (aTc) inducer (Fig. 4b), likely due to leaky expression from pTet[36]. In contrast, when either MCP-(5aa)-SoxS$_{R93A}$ alone or all three components of the CRISPRa system are controlled by pTet, we observed no leaky GFP expression and pTet-inducible GFP levels comparable to that observed with constitutive expression (Fig. 4b). GFP levels for pTet-inductions were measured in late stationary phase cultures; in early stationary phase cultures we observed modestly weaker overall GFP induction compared to that observed with constitutive promoters.

We also tested arabinose-inducible pBAD promoters as an alternative to pTet. With pBAD controlling all three components of the CRISPRa system, we observed inducible GFP expression, but at a level twofold weaker than that observed with constitutive expression. Alternatively, with pBAD controlling both dCas9 and the 1x MS2 scRNA, and constitutive expression of MCP-(5aa)-SoxS$_{R93A}$, we observed strong inducible GFP expression with no significant leaky expression in the absence of arabinose (Fig. 4b). These results were obtained with cultures in early stationary phase, unlike the results with pTet which were obtained in late stationary phase. In preliminary experiments with arabinose-inducible CRISPRi, we observed leaky repression in late stationary phase cultures, possibly because glucose depletion can relieve catabolite repression of the pBAD promoter[37]. We therefore performed the arabinose-inducible experiments in early stationary phase cultures, where no leaky CRISPRi repression was observed. Taken together, these results indicate that effective inducible control of CRISPRa can be achieved with pTet controlling MCP-(5aa)-SoxS$_{R93A}$, or with pBAD controlling dCas9 and the 1x MS2 scRNA.

With inducible control over CRISPRa and CRISPRi, we can implement a two-gene switch from ON/OFF to OFF/ON. We used separate pTet promoters to control all components of the system: dCas9, MCP-(5aa)-SoxS$_{R93A}$, the 1x MS2 scRNA for GFP, and the gRNA for RFP. Upon addition of aTc, GFP expression increases while RFP expression decreases (Fig. 4c). Alternatively, with pBAD promoters controlling dCas9, the 1x MS2 scRNA for GFP, and the gRNA for RFP, and constitutively expressed MCP-(5aa)-SoxS$_{R93A}$, we could also inducibly switch GFP on and RFP off (Fig. 4c).

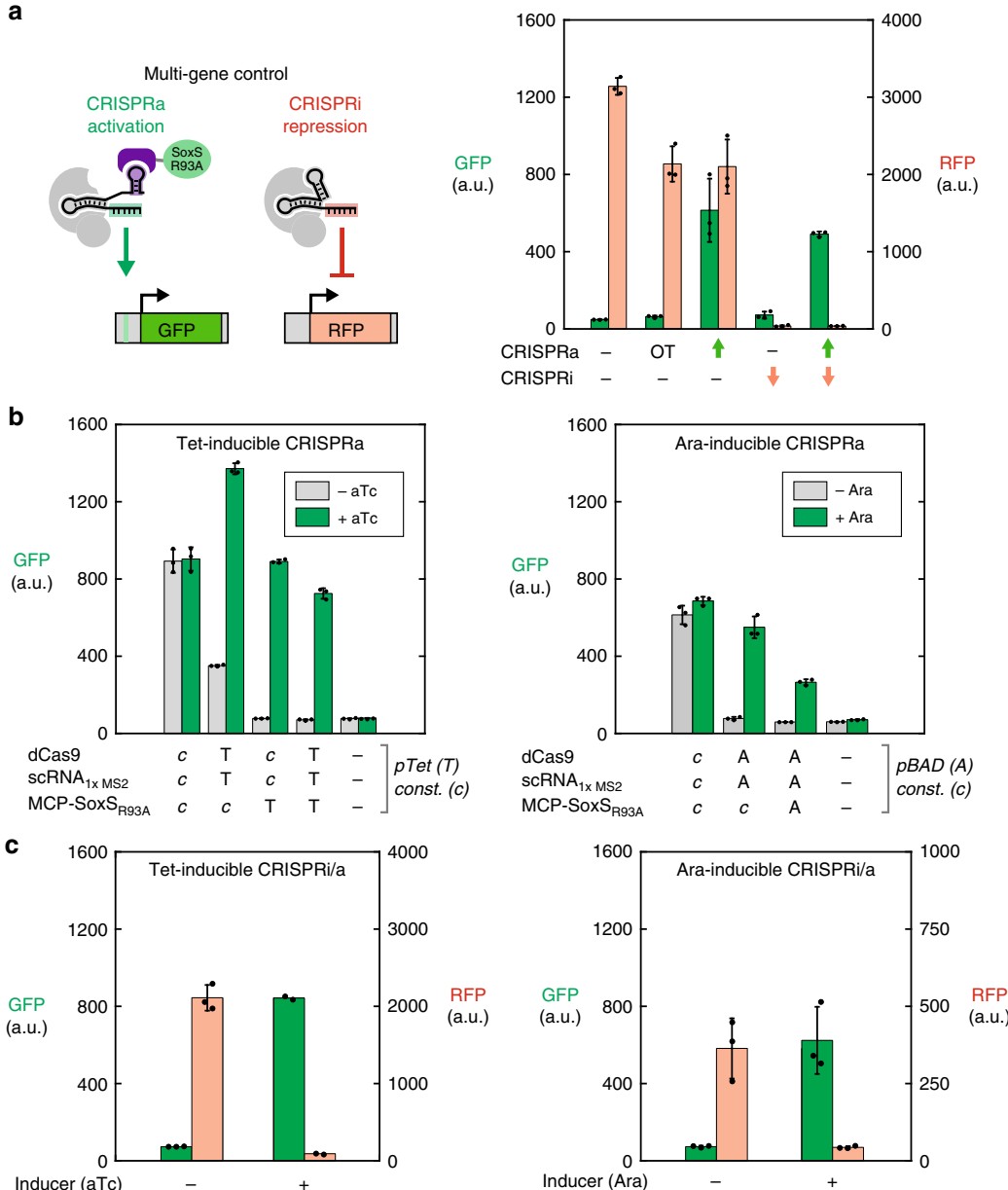

**Fig. 4** Multi-gene inducible control. **a** Expressing multiple gRNAs allows simultaneous regulation of multiple genes. Using an *E. coli* strain with integrated GFP and RFP reporters, a 1x MS2 scRNA targets GFP for activation by recruiting MCP-(5aa)-SoxS$_{R93A}$, while an unmodified gRNA targets RFP for repression. When both gRNAs are expressed, GFP expression increases while RFP expression decreases, and the observed effects are similar to those observed when each gRNA is expressed alone. OT indicates an off-target control for the J106 target site (Supplementary Table 2), which is not present in this strain. Values reported are GFP fluorescence levels measured by flow cytometry. Values are median±s.d. for three biological replicates (specific values are indicated by black dots). **b** CRISPRa can be inducibly controlled with pTet or pBAD (Ara) promoters. In each case, different components of the CRISPRa system (i.e. dCas9, the 1x MS2 scRNA, or MCP-(5aa)-SoxS$_{R93A}$) are controlled by constitutive (c) or inducible (pTet, T or pBAD, A) promoters. pTet is induced with aTc and pBAD is induced with arabinose. For pTet inductions, cultures were harvested after overnight growth (late stationary phase), while for arabinose inductions cultures were harvested after 6 h (early stationary phase). Values reported are GFP or RFP fluorescence levels measured by flow cytometry. Values are median±s.d. for three biological replicates (specific values are indicated by black dots). **c** CRISPRa and CRISPRi can be simultaneously induced with pTet or pBAD (Ara) promoters. For pTet-induction, all components of the CRISPR system are controlled by pTet. For arabinose induction the pBAD promoter controls dCas9 and the guide RNAs, while MCP-(5aa)-SoxS$_{R93A}$ is constitutively expressed. The RFP axis in the Ara panel is smaller than all other RFP axes in this figure because we observed consistently lower fluorescent protein expression in early stationary phase cultures versus late stationary phase cultures, even in the absence of CRISPRi. Values reported are GFP or RFP fluorescence levels measured by flow cytometry. Values are median±s.d. for three biological replicates, except for the +aTc tet inductions, for which two biological replicates were obtained (specific values are indicated by black dots)

**Bacterial CRISPRa is highly sensitive to gRNA target site**. In prior reports of bacterial CRISPRa with dCas9-RpoZ, reporter gene expression depended on the distance of the gRNA target from the TSS, with peak expression occurring at a site located ~90 bases upstream[7]. To determine if a similar dependence applies to different activators recruited via 1x MS2 scRNA.b2 to the CRISPR-Cas complex, we designed a synthetic promoter (J1) that has gRNA target sites with the necessary PAM sequences every 10 bases on both strands upstream of a weak BBa_J23117 promoter driving an RFP reporter gene (Fig. 5a). With the MCP-(5aa)-SoxS$_{R93A}$ activator, we observed a sharp dependence of RFP expression on target site position, with significant RFP expression occurring for sites at 80–90 bases upstream of the TSS on the non-template strand and at 60–80 bases upstream on the template strand (Fig. 5b). To determine if activators with weak activity in our initial assays (Fig. 2a) might be more effective at

different target sites, we tested MCP fusion proteins of TetD, λcII, and αNTD with this promoter. We observed peak activities for these activators at target site positions similar to the most effective sites for SoxS (Fig. 5b), and in no case did we observe dramatic increases in activity relative to our initial assays (Fig. 2a). Similarly, we tested whether an alternative gRNA design (sgRNA 2.0)[11] with MS2 sites embedded within internal hairpins in the gRNA and the MCP-(5aa)-SoxS$_{R93A}$ activator would be more effective at different target sites; we observed no detectable CRISPRa activity with this gRNA design at any target site. Finally, we tested an alternative RNA hairpin (1x PP7 scRNA.b1), that recruits the PP7 coat protein (PCP) fused to SoxS$_{R93A}$. MCP and PCP are structurally homologous proteins, but the MS2 and PP7 RNA hairpins differ significantly and interact in distinct orientations with their cognate binding proteins[38]. Thus, we do not expect activators recruited by MS2 or PP7 to be positioned in

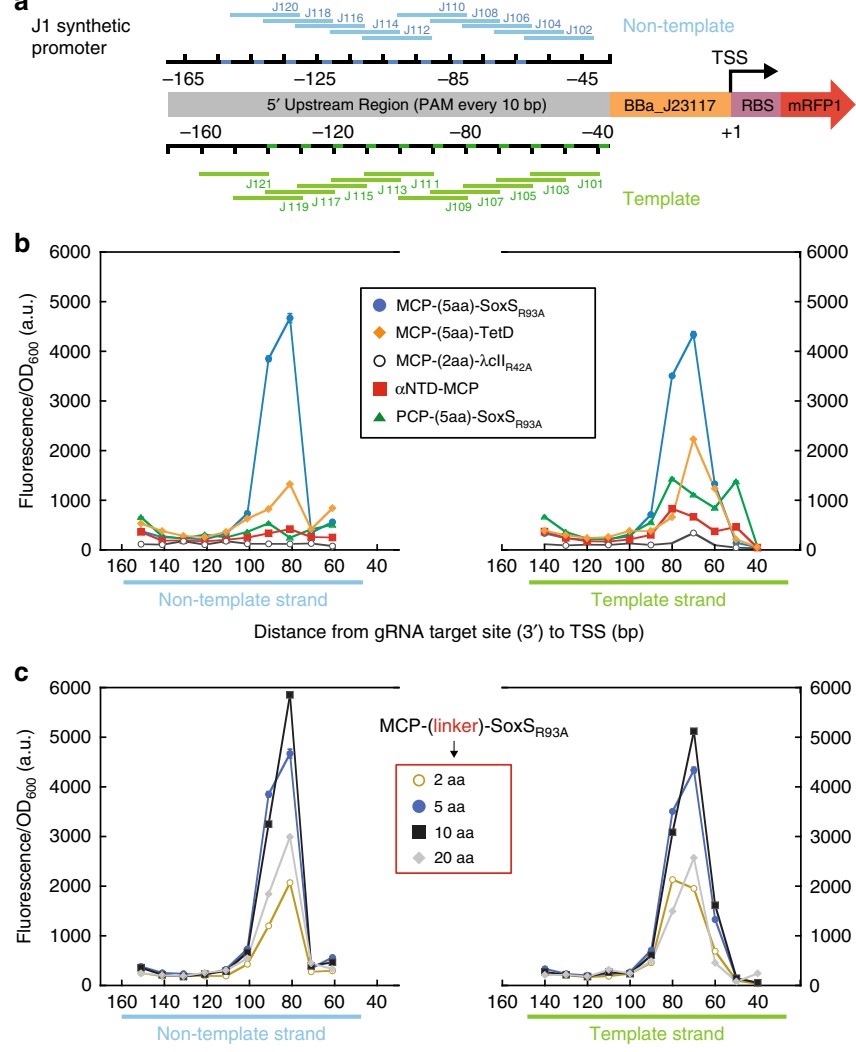

**Fig. 5** Gene activation is highly sensitive to CRISPR target site position. **a** The J1 synthetic promoter has potential gRNA target sites with an appropriately positioned PAM sequence every 10 bases on both strands upstream of a weak BBa_J23117 minimal promoter driving an RFP reporter gene. **b** A plot of RFP expression level versus target site position indicates a narrow region for effective CRISPRa from −80 to −90 upstream of the TSS on the non-template strand and from −50 to −80 on the template strand. The plot shows CRISPRa with several different activators recruited by a 1x MS2 scRNA.b2, including MCP-(5aa)-SoxS$_{R93A}$, MCP-(5aa)-TetD (see Supplementary Fig. 2d), MCP-(2aa)-λcII$_{R42A}$ (see Supplementary Fig. 2e), and αNTD-MCP. We also used a 1x PP7 scRNA.b1 to recruit PCP-(5aa)-SoxS$_{R93A}$. **c** Increasing the linker length between MCP and SoxS$_{R93A}$ does not extend the range of gRNA target sites that are effective for CRISPRa. Maximal activity is observed with 5 and 10 amino acid linkers, while both 2 and 20 amino acid linkers show reduced activity but no significant difference in the position of effective target sites. Values reported are RFP fluorescence levels normalized by the cell density (fluorescence/OD$_{600}$) measured in a plate reader. Values are mean±s.d. for at least three biological replicates

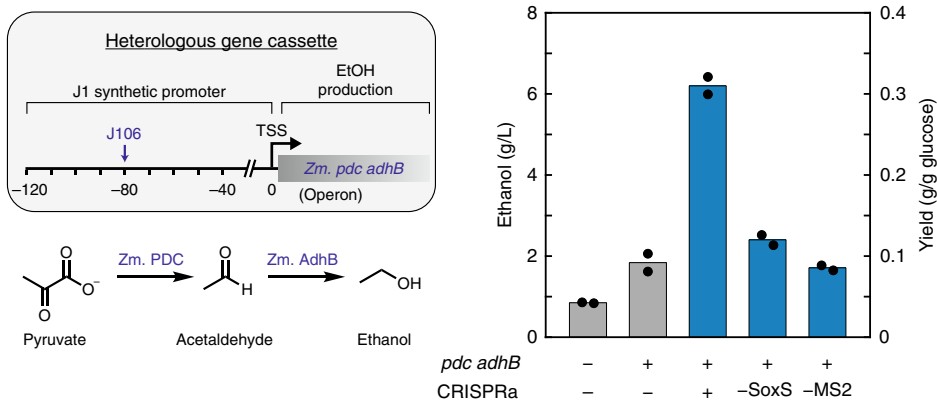

**Fig. 6** CRISPRa-mediated ethanol production. A heterologous gene cassette for pyruvate decarboxylase (*pdc*) and alcohol dehydrogenase (*adhB*) from *Z. mobilis* converts pyruvate to ethanol in *E. coli*. The gene cassette is controlled by a weak promoter (see Supplementary Methods). When a CRISPRa complex (dCas9, MCP-(5aa)-SoxS$_{R93A}$, and 1x MS2 scRNA.b1) is targeted to the J106 site upstream of the promoter, ethanol production increases ~3-fold. In the absence of MCP-(5aa)-SoxS$_{R93A}$ (-SoxS) or with a gRNA lacking the MS2 hairpin (-MS2), ethanol levels are indistinguishable from background strain containing just the *pdc adhB* gene cassette. The left *y*-axis is ethanol titer (g/L) and the right *y*-axis is the same data represented as yield (g ethanol/g glucose). Yield is calculated relative to the initial 2% glucose (20 g/L). Values are the mean for two biological replicates (specific values are indicated by black dots)

similar orientations when recruited to the CRISPR-Cas complex. Perhaps because of this structural difference, PCP-(5aa)-SoxS$_{R93A}$ behaved very differently than MCP-(5aa)-SoxS$_{R93A}$. The PCP construct was relatively ineffective at all sites on the non-template strand and moderately effective on the template strand (Fig. 5b). These results suggest that bacterial CRISPRa is highly sensitive to target position upstream of the TSS, and that it may be possible to find different combinations of activators, recruitment domains, and target sites that perform better than the most effective SoxS constructs tested here. Because all of the activators tested here interact directly with RNA polymerase, albeit at different protein interfaces, we suggest that strong activation requires a CRISPR target site that allows RNA polymerase to bind to the minimal promoter (i.e. the −35 and −10 sites) and the activator simultaneously, which puts a significant constraint on the upstream target sites that can be used for CRISPRa.

To extend the range of effective target sites, we hypothesized that a longer, flexible linker between MCP and SoxS$_{R93A}$ might allow activators to recruit RNA polymerase to the promoter from a broader range of target sites. We therefore tested linkers with 2, 5, 10, or 20 amino acids. Extending the original 5 amino acid linker to 20 amino acids should lengthen the linker by ~57 Å (assuming a worm-like chain model for a flexible peptide linker with 3.8 Å/residue)[39]. Because target sites on the J1 promoter are spaced ~33 Å apart (10 bp spacing with 3.3 Å/bp for B-DNA), we expect that the extended linker should broaden the range of target sites by at least 10–20 bp in either direction from the optimal range observed with the 5 amino acid linker. Surprisingly, we observed no significant broadening of the effective target site range with longer linkers (Fig. 5c). We observed maximal activity with 5 and 10 amino acid linkers, while both 2 and 20 amino acid linkers show reduced activity but no significant difference in the position of effective target sites. While it remains to be seen if it is possible to design a bacterial CRISPRa complex that functions over a broader target range, simply extending linker lengths between MCP and the SoxS$_{R93A}$ activator was not effective.

**Activating a metabolic gene cluster for ethanol biosynthesis**. An important practical application of bacterial CRISPRa will be to enable complex and dynamic multi-gene circuits to control biosynthetic pathways. The capability to express pathway enzymes while repressing competing enzymes may lead to

significant improvements in product yields[40]. As a proof of concept, we tested whether we could upregulate a heterologous *pdc adhB* gene cassette from *Zymomonas mobilis*, which converts pyruvate to ethanol and has previously been used for ethanol production in *E. coli* (Fig. 6)[41,42]. When we targeted the heterologous gene cassette with our CRISPRa system, we observed a threefold increase in ethanol production relative to cells without CRISPRa (Fig. 6). This result suggests that CRISPRa-based transcriptional control can be used to activate biosynthetic pathways. We expect that combining CRISPRa-based control of heterologous biosynthetic genes with CRISPRi on competing endogenous genes will enable rapid explorations of a large space of genetic circuit architectures to improve biosynthesis yields, and further improvements may be realized by targeting additional metabolic genes for activation or repression, and by coupling the system to dynamically-regulated or inducible promoters.

## Discussion

In this work, we have identified multiple synthetic transcriptional activators compatible with CRISPRa in *E. coli*, including SoxS, TetD, and AsiA. The most effective activator, SoxS, is substantially stronger than the previously reported RpoZ-based synthetic activator and does not require additional host-strain modifications for function. SoxS may be an effective activator in part because it interacts with the C-terminal domain of RpoA, which is connected to RNA polymerase by a relatively flexible tether[12,43]. Using SoxS and other activators, we find a surprisingly sharp dependence on the CRISPR target site distance from the TSS for effective gene activation, with the optimal gRNA sites positioned within a narrow window roughly 60–90 bases upstream of the transcription start site of a heterologous reporter gene. This range is strikingly smaller than the broad target site range observed for effective CRISPRa in eukaryotic cells, where many target sites in the 1–500 base range upstream of the TSS are effective[44]. One notable difference between bacteria and eukaryotes is that the effective bacterial activators identified in this and other work all appear to directly interact with RNA polymerase, while in eukaryotic cells transcription factors may act over longer distances via indirect chromatin modifications. Nevertheless, even given these constraints the ability to perform effective CRISPRa in bacteria will open significant avenues for engineering bacterial systems. Most importantly, we expect that

controlling branch points in metabolic networks with simultaneous gene activation and repression will improve biosynthetic yields beyond that obtained simply by constitutively overexpressing heterologous pathways. Further, we can use inducible promoters to control the timing of gene expression, and we can build more sophisticated dynamic gene expression programs using protein or RNA biosensors to sense and respond to cellular metabolic states[45,46].

Two future challenges remain for broad applicability of CRISPRa in bacteria. First, it will be necessary to identify predictive rules for targeting and activating gene expression at endogenous sites. In preliminary experiments, we have observed only modest increases in gene expression at endogenous sites, and the position of effective target sites does not necessarily correspond to the optimal target sites that we observed for heterologous promoter activation. It is possible that our observed distance dependence is not generalizable to different promoters, that endogenous regulatory factors interfere with CRISPRa, or that different types of promoters require distinct transcriptional activation domains. In practice, if predictive rules for targeting arbitrary endogenous genes remain elusive, we envision using gene editing to introduce heterologous promoters at target sites of interest, which will enable us to regulate endogenous genes as part of a multi-gene CRISPRi/a control program.

A second outstanding challenge for bacterial CRISPRa is to develop a system that is portable across bacterial species of high commercial and industrial value, such as strains that have the ability to utilize carbon sources like $CO_2$, CO, methane, or lignocellulose, and alternative energy sources such as $H_2$ or light[1,2]. For gene repression, CRISPRi has been successfully used in a broad range of bacterial species[16,47–50], suggesting that the programmable DNA targeting component of the system is portable. For CRISPRa, many of our candidate gene activators interact with motifs on bacterial RNA polymerase that are highly conserved across a broad range of bacteria[51], including several that are relevant for industrial biosynthesis or microbiome engineering. In particular, SoxS interacts with a surface on the RNA polymerase α subunit that is well-conserved in gammaproteobacteria, alphaproteobacteria, bacteroides, gram-positive bacteria, and even to a lesser extent in cyanobacteria (Supplementary Fig. 8). This conserved interface may allow CRISPRa systems developed in *E. coli* to be ported to non-model bacteria with a wide range of useful biological functions.

## Methods

**Bacterial strain construction and manipulation.** Plasmid constructs were cloned and *E. coli* cells were cultured using standard molecular biology methods. Gene knockouts and integrations of fluorescent reporters were performed by recombineering in a strain with a genomically-integrated lambda red system under the control of a temperature-sensitive promoter (NM700)[52–54]. Modified genome fragments were then transferred to MG1655 by P1 transduction[55]. The sfGFP reporter was integrated at the *nfsA* locus. The mRFP reporter (for CRISPRi experiments) was integrated at *rbsAR*[56]. The ΔrpoZ strain was constructed by recombineering a pKD13-derived KanR linear PCR cassette with 36 base homology overhangs to flanking sites at the rpoZ locus. The ΔrpoZ::KanR knockout genomic fragment was transferred to MG1655 by P1 transduction, and the FRT-flanked KanR cassette was eliminated by transformation with the pCP20 helper plasmid to express FLP recombinase, followed by incubation at 42 °C to cure the plasmid. CRISPR-Cas system components were delivered on plasmids as described in Supplementary Table 3. *S. pyogenes* dCas9 was expressed from its endogenous *S. pyogenes* Cas9 promoter (cloned from pWJ66,[7] addgene #46570). Candidate effector proteins fused to RBPs (i.e. MCP and PCP) were expressed with the medium-strength BBa_J23107 promoter. Guide RNAs were expressed from the strong BBa_J23119 promoter[6]. BBa sequences are available from the Repository of Standard Biological Parts (http://parts.igem.org). zwf-LacZ and fumC-LacZ reporter genes were constructed following previously described designs (see Supplementary Methods)[57,58]. Complete annotated sequences of the reporter genes and CRISPR-Cas system components are included in the Supplementary Methods along with a list the engineered *E. coli* strains (Supplementary Table 1). Guide RNA target sites are listed in Supplementary Table 2.

**Flow cytometry.** Cells were inoculated in EZ-RDM (Teknova) supplemented with appropriate antibiotics and grown in 96-deep-well plates at 37 °C, 220 RPM overnight. Late stationary phase cultures were then diluted 1:40 in PBS and analyzed on a MACSQuant VYB flow cytometer (Miltenyi Biotec). To enrich for single cells, a side scatter threshold trigger (SSC-H) was applied. To gate for single bacterial cells, we first selected events along the diagonal of the SSC-H vs. SSC-A plot[59]. We then excluded events that appeared on the edges of the SSC-A vs. FSC-A plot, and events that appeared on the edge of the fluorescence histogram (Supplementary Fig. 9).

For inducible CRISPR system construction with pTet or pBAD promoters, strains were inoculated in 3 mL LB medium supplemented with antibiotics and grown overnight at 37 °C, 220 RPM. On the next day, they were diluted 1:100 in 500 μL EZ-RDM supplemented with antibiotics and induced with 1 μM anhydrotetracycline (aTc) for pTet or 100 mM L-arabinose for pBAD. Non-induced controls were prepared for each strain. For pTet, cells were grown at 37 °C, 220 RPM overnight. For pBAD, cells were grown at 37 °C, 220 RPM for 6 h. In both cases, fluorescence was assayed via flow cytometry as described above.

**LacZ reporter assays.** LacZ reporter assays for SoxS activation of *zwf* and *fumC* promoters were performed following a previously reported protocol with minor modifications[60,61]. Specifically, cultures were grown for 18 h in EZ-RDM supplemented with antibiotics. Absorbance at 600 nm ($OD_{600}$) was measured from 150 μL samples in a Biotek Synergy HTX plate reader. 20 μL aliquot from each culture were added to 80 μL permeabilization solution [100 mM $Na_2HPO_4$, 20 mM KCl, 2 mM $MgSO_4$, 0.8 mg/mL hexadecyltrimethylammonium bromide, 0.4 mg/mL sodium deoxycholate, 5.4 μL/mL β-mercaptoethanol]. Samples were incubated at 30 °C for 20 min. To initiate the reaction, 600 μL of substrate solution [60 mM $Na_2HPO_4$, 40 mM $NaH_2PO_4$, 1 mg/mL o-nitrophenyl-β-D-galactopyranoside, 2.7 μL/mL β-mercaptoethanol] was added. Samples were incubated at 30 °C until visible color developed, at which point 700 μL of stop solution [1 M $Na_2CO_3$] was added and the reaction time was recorded. Samples were centrifuged for 10 min to pellet cell debris, and the supernatants were removed. Absorbance at 420 nm ($A_{420}$) was measured from 150 μL samples in a plate reader. LacZ activity was calculated according to the formula: Miller Units = $(1000 \times A_{420})/(OD_{600} \times 0.02$ mL $\times$ time).

**J1 reporter sequence design.** Custom Python scripts were used to generate a tiling array containing an NGG PAM site every 10 nucleotides on each strand according to the following base unit: 5′-NNNCCNNNGG-3′. 10000 sequences of length 500 nt were generated by randomly sampling N nucleotides, adjusted for endogenous GC content in *E. coli* (BNID '100528 [http://bionumbers.hms.harvard.edu/bionumber.aspx?id=100528]')[62]. We discarded any sequences with four consecutive identical nucleotides. Sequences containing known transcription factor binding sites (using the experimental TF binding site dataset from RegulonDB)[63] were also discarded. Of the remaining sequences, we arbitrarily chose one, labeled as J1, and confirmed that it had no detectable homology to the *E. coli* genome. We placed a 170 bp fragment of this sequence upstream of the weak BBa_J23117 constitutive promoter (http://parts.igem.org).

**Plate reader experiments.** For fluorescent reporter experiments with the J1 reporter, cells were inoculated in 2 mL EZ-RDM supplemented with appropriate antibiotics and grown at 37 °C, 220 RPM overnight. $OD_{600}$ and observed fluorescence values were measured in a Biotek Synergy HTX plate reader using 150 μL of the overnight culture in flat, clear-bottomed 96-well plates (Corning). For mRFP detection, the excitation wavelength was 540 nm and the emission wavelength was 600 nm.

**Ethanol fermentations.** *E. coli* MG1655 cells were transformed with the pCD355 plasmid containing the *pdc adhB* gene cassette from *Z. mobilis* under the control of a weak promoter (Supplementary Table 3) and with or without components of the CRISPRa system. Transformed cells were inoculated in 5 mL LB (2% glucose) supplemented with appropriate antibiotics and grown overnight at 37 °C, 220 RPM. $OD_{600}$ measurements were taken from the overnight cultures, and 2.5 OD·mL of the culture was diluted into 50 mL LB supplemented with antibiotics. Cultures were grown aerobically for 18 h at 37 °C, 220 RPM. $OD_{600}$ measurements were taken, and 150 OD·mL was pelleted and resuspended into 25 mL M9 media with 2% dextrose and antibiotics in a 125 mL conical flask. Flasks were sealed with rubber septa with a needle connected to a balloon to relieve pressure from $CO_2$ production. Cultures were incubated at 37 °C without shaking for 4 days. Ethanol concentrations in supernatants were measured with an Ethanol assay kit (R-Biopharm). We counted viable cells by plating on selective media before and after fermentation and observed no decrease in viable cells after the fermentation. To evaluate plasmid stability over the course of the fermentation, we performed minipreps on cells after the fermentation. Analytical restriction digests indicated no large-scale recombination events, and no point mutations were detected by sequencing.

**Quantitative RT-PCR.** Strains were inoculated in 5 mL LB supplemented with appropriate antibiotics and grown overnight at 37 °C, 220 RPM. Cultures were then

diluted 1:100 in 5 mL EZ-RDM supplemented with antibiotics and grown to $OD_{600}$ 0.5 (using 150 μL samples in flat clear bottomed 96-well plates in a Biotek Synergy HTX plate reader). Cultures were pelleted, flash-frozen in liquid nitrogen, and stored at −80 °C. Total RNA was extracted using an Aurum Total RNA Mini Kit (Bio-Rad). 1 μg of RNA was converted to cDNA using iScript reverse transcriptase (Bio-Rad) in 20 μL reactions. qPCR was performed using SsoAdvanced Universal SYBR Green Supermix (Bio-Rad) with a 58 °C annealing temperature and 10 μL reaction volumes. qPCR reactions were performed in triplicate on a CFX Connect (Bio-Rad) using 0.5 ng of cDNA, 400 nM primer concentration, and 15 s extension time. 16 S rRNA was used for normalization. Control samples without a template and without reverse transcriptase were analyzed to confirm gene-specific amplification and absence of gDNA contamination. Primer sequences are listed in Supplementary Table 4. Expression levels for each gene were calculated relative to 16 S using the $\Delta\Delta C_T$ method[64].

**Code availability**. Custom Python code to generate tiling arrays with PAM sequences (detailed in "J1 reporter sequence design" section of Methods) is available upon request.

**Data availability**. All data from this study are available upon request.

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

## Acknowledgements

We thank Maureen Thomason, Mary Lidstrom, Frances Chu, Willy Voje, Jason Stevens, Chuhern Hwang, and members of the Zalatan and Carothers groups for technical assistance, advice, and helpful discussions. The NM700 strain was a gift from Nadim Majdalani and Susan Gottesman. This work was supported by a Career Award at the Scientific Interface from the Burroughs Wellcome Fund (J.G.Z.), an NSF Award MCB 1517052 (J.M.C.), and a University of Washington Presidential Innovation Award (J.M.C.).

## Author contributions

J.G.Z. conceived the project. C.D., J.F., J.M.C., and J.G.Z. designed experiments, analyzed data, and wrote the manuscript. C.D., J.F., and A.P. performed experiments.

## Additional information

**Competing interests:** The authors declare no competing interests.

