## [Peer Review File · Nature Communications]

Reviewers' comments:

Reviewer #1 (Remarks to the Author):

Building on previous work, this well done study demonstrates the use of the CRISPR-Cas system to effect simultaneous gene activation and gene repression in *E. coli*. Importantly, the authors have re-engineered a previously described system for gene activation, making it significantly more robust. The data are clearly presented and compelling, and the analysis is impressively comprehensive. The findings should greatly facilitate efforts to engineer synthetic gene expression programs in *E. coli* and potentially other bacteria as well. The study thus represents a significant contribution to the field.

Specific comments:

1. pg 3: second paragraph of Introduction, lines 5-6 and lines 6-8: Reference 14 should be cited along with reference 6 when introducing CRISPRi and should similarly be cited when introducing CRISPRa.
2. pg 3, second paragraph of Introduction, lines 8-11: This description of engineered transcription activation in bacteria is confusing as written. In particular, the reference to transcription activation domains is misleading. The cited studies show that any protein-protein interaction can bring about transcription activation provided that one of the interacting proteins is fused to a subunit of RNA polymerase (RNAP) and the other is fused to a DNA-binding domain/protein with the ability to bind a DNA recognition site suitably positioned upstream of the target gene. Alternatively, transcription activation can be achieved by appending a subunit of RNAP (or an RNAP-binding protein) directly to the DNA-binding domain. Accordingly, RpoZ should be identified as a subunit of RNAP. Although the authors distinguish between subunits of RNAP and transcription regulators in the Results section, a clarified description of how activation is achieved would make the paper more accessible to a broad readership.
3. pg 4, lines 6-8: The findings reported in reference 14 also indicated that CRISPRa in *E. coli* is highly sensitive to the location of the gRNA target site.
4. Figure 1B: The use of orange and blue rectangles in this figure is confusing. What distinguishes the two orange rectangles and the two blue rectangles? Why is only one example of each targeted?
5. pg 6, end of first paragraph: It would be helpful to provide a few more words about how SoxS normally functions to activate transcription, as this has not been described.
6. pg 8, middle paragraph: Do the authors have an explanation for the leaky expression that was observed when MCP-(5aa)-SoxS-R93A was produced constitutively? This finding seems to be at odds with the results shown in Fig 3E (unless the tet promoter is itself leaky).
7. pg 9, 7 lines up from bottom: For MCP-(5aa)-SoxS-R93A, it looks as if significant RFP expression occurs for sites at 60-80 bases upstream of the TSS (not 50-80).
8. pg 12, Discussion first sentence: For clarity, the sentence could be reworded along the following lines: "... one of which, based on the transcription regulator SoxS, is substantially stronger than the previously reported RpoZ-based synthetic activator..."
9. Discussion section: A possible explanation for the fact that SoxS functions as a more potent synthetic activator in this system than RpoZ is that SoxS interacts with the alpha CTD, which is flexibly tethered to the body of RNAP and presents a particularly accessible target at the back end of RNAP.

10. pg 13, lines 9-13: Because SoxS appears to interact with an extended surface of the alpha CTD (including the three key residues at positions 261, 263 and 265, alanine substitutions of at least 7 residues affect the SoxS-alpha CTD interaction either negatively or positively), the likelihood that SoxS can be used to enable CRISPRa in distantly related bacteria seems somewhat low.

Reviewer #2 (Remarks to the Author):

In this manuscript, Dong et al. sought to develop synthetic transcription factors in bacteria, which could be coupled to programmable DNA binding domains and controlled by inducible promoters to engineer responsive multi-gene expression programs. Very few transcriptional activation domains have been reported to be effective when fused to modular DNA binding domains in bacteria. Using SoxS, which can interact with a highly conserved site on RNA polymerase, they could activate one target gene while simultaneously repressing a different target gene with CRISPRi. They also show that bacterial CRISPRa can be used to increase the output of a heterologous ethanol biosynthesis pathway in *E. coli*. Of interest, they also noted that gene activation in *E. coli* is highly sensitive to the location of the gRNA target site. Finally, they claim that their CRISPRa toolkit may be portable to a broad range of bacterial species, but this was not shown. Overall, I enjoy reading the manuscript.

1. First and foremost, line and page numbering systems would have been really appreciated.
2. Page 6 line 11, what are the typical genes normally regulated by SoxS? Have you checked some of them to see if there was an impact? Perhaps this could be discussed.
3. Page 6 line 14: Minimize off-target effect? What do you mean? How was this tested? May be missing something here.
4. Page 6 line 19: You hypothesized that it is due to gene activation/increased expression of full length scRNA but was this measured? Could this be due somehow to RNA stability?
5. Page 7 line 3: off-target effects again. Please explain.
6. Page 7 line 13: "...we plotted CRISPRa versus endogenous promoter activity...". According to the legend of Figure 3, these promoter activities are from the literature. While I don't necessarily have an issue with this, I am somewhat surprised that they were not tested in your conditions?
7. Page 7 line 22: The comparison between Figure 3E and Figure 2B, the data should be on the same panel.
8. Page 8 line 8: The sentence is a bit misleading as according to Figure 4A the constructs are different. The CRISPRi repression construct should be explained to some extent.
9. Page 8 line 23: Unless I missed understood the MM section, these were overnight cultures (18 hrs?). Would this be considered very late stationary phase? In your bacterial growth assays (Fig. S5) you seem to reach the stationary phase already after 5-6 hours?
10. Page 8 line 23: Exponential phase means what in terms of hours compared to the "stationary" cultures? Perhaps you should show the data (supp mat) for the exponential phase? Why did you use such "old cultures" to measure GFP levels in these induced cells?
11. Page 8 Line 25: constitutive.
12. Page 9 line 1: What late exponential means in terms of hours? 6 hours? I would say early stationary based on Fig S5.
13. Page 9 Line 3: Stationary cultures as overnight cultures? Any reason why the leakiness in these conditions?
14. Page 9 line 25: I do not understand why these regulators were retested here. And why only these three? Please explain. Perhaps these results should also be in supp mat.
15. Page 9 line 27: I also do not understand why this 2.0 construct was tested at this stage as well as the construct itself. Perhaps these results should also be in supp mat.
16. Page 10 line: While I understand the rationale, is there a risk that SoxS may not be the best regulators (among the 12 tested) with MS2 MCP?

17. Page 11 line 1: *Z. mobilis* (*Zymomonas*) should be defined.
18. Page 11 lines 5-6: Were the growth curves done with the ethanol construct? If not, perhaps this should be moved to the appropriate section. Again the ethanol experiments were conducted for several days (!) while the growth curves for 6 hours. Cells counts should have been performed at the end of the experiments.
19. Page 12 line 1: Have you really identified multiple transcriptional activators? One seems to be more accurate.
20. Page 12 line 24: What endogenous sites were tested? What modest means? What do you mean by preliminary experiments? Tested once?
21. Page 13 line 12: Would this be relatively easy to test? Your best constructs in least another closely-related gram-negative bacterium?
22. What is the long-term stability of the constructs? Have you tested over multiple generations? The ethanol experiments seems to indicate that it is relatively stable (I would assume yes if you keep the selective pressure). Have you resequenced the construct after the ethanol fermentation, which appears to have run for 4 days?
23. Figure 4: The Y axis values for the GFP expression should all be the same to allow better comparison.
24. Figure 5: The Y axis values for the GFP expression should all be the same to allow better comparison.
25. References: Bacterial names are not in italic.
26. References: Title of the cited papers will have to be checked for uppercase / lowercase.

Manuscript: NCOMMS-18-03595

Title: Synthetic CRISPR-Cas Gene Activators for Transcriptional Reprogramming in Bacteria

POINT BY POINT RESPONSE & REVISIONS

Reviewer comments are in *red* and our responses are in black. Changes in the revised manuscript have been highlighted in red.

Detailed Response to Reviewer 1

Reviewer 1 noted that “The data are clearly presented and compelling, and the analysis is impressively comprehensive... The study thus represents a significant contribution to the field.” The reviewer raised several specific comments:

1. pg 3: second paragraph of Introduction, lines 5-6 and lines 6-8: Reference 14 should be cited along with reference 6 when introducing CRISPRi and should similarly be cited when introducing CRISPRa.

We have made the requested change.

2. pg 3, second paragraph of Introduction, lines 8-11: This description of engineered transcription activation in bacteria is confusing as written. In particular, the reference to transcription activation domains is misleading. The cited studies show that any protein-protein interaction can bring about transcription activation provided that one of the interacting proteins is fused to a subunit of RNA polymerase (RNAP) and the other is fused to a DNA-binding domain/protein with the ability to bind a DNA recognition site suitably positioned upstream of the target gene. Alternatively, transcription activation can be achieved by appending a subunit of RNAP (or an RNAP-binding protein) directly to the DNA-binding domain. Accordingly, RpoZ should be identified as a subunit of RNAP. Although the authors distinguish between subunits of RNAP and transcription regulators in the Results section, a clarified description of how activation is achieved would make the paper more accessible to abroad readership.

We have revised this portion of the manuscript as suggested by the reviewer, with revisions underlined here: “In bacteria, however, there are very few transcriptional activation domains that have been reported to be effective when fused to modular DNA binding domains. Bacterial two-hybrid systems have been constructed with pairs of candidate interacting proteins separately fused to RNA polymerase subunits and DNA binding proteins, and it is also possible to fuse RNA polymerase subunits directly to DNA binding domains to activate transcription¹²⁻¹⁴. One of the RNA polymerase subunits, RpoZ, has been coupled to the CRISPR system to activate gene expression^{7,15-18}.”

3. pg 4, lines 6-8: The findings reported in reference 14 also indicated that CRISPRa in E. coli is highly sensitive to the location of the gRNA target site.

We have cited this reference as suggested by the reviewer. We cited this work when describing the result, but we agree that it should also be cited prominently for the same point in the introduction.

4. Figure 1B: The use of orange and blue rectangles in this figure is confusing. What distinguishes the two orange rectangles and the two blue rectangles? Why is only one example of each targeted?

We apologize for the confusion. It appears that the graphic submitted with the original manuscript was incomplete and missing key components that should have been present. We intended for the graphic to show multiple genes being targeted by an inducible input, with each orange rectangle (gene) targeted with a green arrow to indicate activation and each blue rectangle (gene) targeted with a red line to indicate repression. The corrected version is included with the revised manuscript.

5. pg 6, end of first paragraph: It would be helpful to provide a few more words about how SoxS normally functions to activate transcription, as this has not been described.

We have added the following description to the manuscript shortly after describing the initial results with SoxS on pg 5-6: “The most effective activator, SoxS, is a member of the AraC family of transcription factors that activates expression by binding to RNA polymerase in a pre-recruitment complex and scanning the genome to find DNA targets located in promoter regions^{20,28}. This mechanism of action could explain the effectiveness of SoxS as a candidate activator for CRISPRa.”

6. pg 8, middle paragraph: Do the authors have an explanation for the leaky expression that was observed when MCP-(5aa)-SoxS-R93A was produced constitutively? This finding seems to be at odds with the results shown in Fig 3E (unless the tet promoter is itself leaky).

We do suspect that the tet promoter is leaky in this case, and we have added this point to the manuscript. We include a citation to our recently published manuscript on inducible CRISPRi systems that includes data to justify this point.

7. pg 9, 7 lines up from bottom: For MCP-(5aa)-SoxS-R93A, it looks as if significant RFP expression occurs for sites at 60-80 bases upstream of the TSS (not 50-80).

We apologize for the confusion, this was a mistake that is now corrected in the revised manuscript.

8. pg 12, Discussion first sentence: For clarity, the sentence could be reworded along the following lines: “... one of which, based on the transcription regulator SoxS, is substantially stronger than the previously reported RpoZ-based synthetic activator...”

We have made the suggested change.

9. Discussion section: A possible explanation for the fact that SoxS functions as a more potent synthetic activator in this system than RpoZ is that SoxS interacts with the alpha CTD, which is flexibly tethered to the body of RNAP and presents a particularly accessible target at the back end of RNAP.

We thank the reviewer for the suggestion. We have added this point to the manuscript, along with citations to Blatter et al. 1994 and Dove et al. 1997, which support this idea.

10. pg 13, lines 9-13: Because SoxS appears to interact with an extended surface of the alpha CTD (including the three key residues at positions 261, 263 and 265, alanine substitutions of at least 7 residues affect the SoxS-alpha CTD interaction either negatively or positively), the likelihood that SoxS can be used to enable CRISPRa in distantly related bacteria seems somewhat low.

We respectfully disagree, although we acknowledge that the point is arguable and that we currently lack experimental data to test this hypothesis. We are working towards such experiments but view them as beyond the scope of this manuscript. We do find the evidence for a conserved interface compelling enough to justify an attempt. We note that the conserved interface in RNA polymerase extends well beyond the three residues we originally highlighted in Supplementary Figure 8, which were chosen because they had the largest effect on the SoxS- α CTD interaction in prior reports. All seven of the residues identified previously, including the three mentioned above, are well-conserved. We have modified the figure to highlight all of these residues, along with the surrounding sequence regions which are also well-conserved, and modified the figure legend to describe these sites in detail.

Detailed Response to Reviewer 2

Reviewer 2 also had a favorable impression of the manuscript, and provided a number of comments which we address below.

1. First and foremost, line and page numbering systems would have been really appreciated.

We have added page and line numbers to the manuscript, and we apologize for the omission!

2. Page 6 line 11, what are the typical genes normally regulated by SoxS? Have you checked some them to see if there was an impact? Perhaps this could be discussed.

SoxS activates a number of genes in response to oxidative stress. We have now measured SoxS activity at two representative endogenous gene targets, *zwf* and *fumC*, using a LacZ reporter assay. We observe activation of these reporter genes when MCP-SoxS fusion constructs are expressed. Further, we observe that mutations in SoxS that disrupt DNA binding lead to substantial reductions in LacZ activity, consistent with previously reported results. These data imply that we can decouple transcriptional activation from DNA binding using SoxS mutants that retain CRISPRa activity. In our original manuscript, we made this suggestion based on literature data for reporter gene activation by SoxS mutants; we have now directly measured reporter gene activation with MCP-SoxS fusion constructs and confirmed the original literature data. We have added additional discussion to pages 5-7 and data to Figure 3 and Supplementary Figure 3.

3. Page 6 line 14: Minimize off-target effect? What do you mean? How was this tested? May be missing something here.

We apologize for the confusion – we meant for this point to refer specifically to off-target effects at endogenous SoxS gene targets resulting from expression of SoxS constructs. We have clarified this point in the revised manuscript. This point was tested by using mutants of SoxS that disrupt binding to endogenous DNA target sites, and we have conducted additional experiments as described in response to #2 above to justify this point.

4. Page 6 line 19: You hypothesized that it is due to gene activation/increased expression of full length scRNA but was this measured? Could this be due somehow to RNA stability?

The reviewer is correct – we cannot distinguish between increased expression and reduced degradation. We have modified the manuscript to state that the observed effect is “likely due to increased steady-state levels of the full length scRNA.”

5. Page 7 line 3: off-target effects again. Please explain.

We have revised the manuscript as described in #3 above to clarify this point.

6. Page 7 line 13: "...we plotted CRISPRa versus endogenous promoter activity...". According to the legend of Figure 3, these promoter activities are from the literature. While I don't necessary have an issue with this, I am somewhat surprised that they were not tested in your conditions?

We have now performed these experiments, as described in response to #2 above. The results generally agree with the trends in the literature data that we cited in our original manuscript.

7. Page 7 line 22: The comparison between Figure 3E and Figure 2B, the data should be on the same panel.

We have repeated the experiment with all samples measured at the same time and we observe the same trends. The new data is combined in the same panel and presented in new Supplementary Figure 4. We moved this panel to Supplemental because Fig. 3D has been expanded with new data (see #2 above).

8. Page 8 line 8: The sentence is a bit misleading as according to Figure 4A the constructs are different. The CRISPRi repression construct should be explained to some extent.

We have modified the relevant sentences to avoid the ambiguity with the use of "gRNA".

9. Page 8 line 23: Unless I missed understood the MM section, these were overnight cultures (18 hrs?). Would this be considered very late stationary phase? In your bacterial growth assays (Fig. S5) you seem to reach the stationary phase already after 5-6 hours?

The reviewer is correct that these cultures should be referred to as late stationary phase. We have updated the manuscript here and elsewhere to consistently refer to overnight cultures as late stationary, 5-6 hour cultures as early stationary phase, and 2-3 hours as exponential phase.

10. Page 8 line 23: Exponential phase means what in terms of hours compared to the "stationary" cultures? Perhaps you should show the data (supp mat) for the exponential phase? Why did you use such "old cultures" to measure GFP levels in these induced cells?

Please see response to #9 above. We observe similar trends in gene activation in exponential and stationary phase, with larger overall magnitudes observed in stationary phase. We have clarified this point in the legend of Figure 1, in the text on pages 7-8, and in data presented in Supplementary Figure 6.

11. Page 8 Line 25: constitutive.

We have corrected the typo.

12. Page 9 line 1: What late exponential means in terms of hours? 6 hours? I would say early stationary based on Fig S5.

Please see the response to #9 above.

13. Page 9 Line 3: Stationary cultures as overnight cultures? Any reason why the leakiness in these conditions?

We have clarified the description to “late stationary phase”, as described in response to #9 above. We suspect pBAD may be leaking because glucose depletion in late stationary phase could relieve catabolite repression of the pBAD promoter (Guzman et al., 1995, PMID 7608087). We have included this statement in the revised manuscript.

14. Page 9 line 25: I do not understand why these regulators were retested here. And why only these three? Please explain. Perhaps these results should also be in supp mat.

We chose these regulators to determine if activators that showed weak but detectable activity in our initial assays (Fig. 2A) might be more effective at different target sites. We have modified the text to clarify this point.

15. Page 9 line 27: I also do not understand why this 2.0 construct was tested at this stage as well as the construct itself. Perhaps these results should also be in supp mat.

Similar to #14 above, we tested whether the sgRNA 2.0 construct might show significant activity at alternative target sites, since it is well-established as a highly effective activation platform in eukaryotic cells. Initial data for this construct was presented in Supplementary Fig. 2C.

16. Page 10 line: While I understand the rational, is there a risk that SoxS may not be the best regulators (among the 12 tested) with MS2 MCP?

We agree that it is possible that there is a combination of target site and regulator that might perform better than the sites tested so far with either MS2/MCP or PP7/PCP. We have added a sentence to the manuscript on page 10 to acknowledge this possibility.

17. Page 11 line 1: Z. mobilis (Zymomonas) should be defined.

We have included the full name of the strain as requested.

18. Page 11 lines 5-6: Were the growth curves done with the ethanol construct? If not, perhaps this should be moved to the appropriate section. Again the ethanol experiments were conducted for several days (!) while the growth curves for 6 hours. Cells counts should have been performed at the end of the experiments.

The growth curves were not done with the ethanol construct. We have moved this data to the “Optimizing Activity of the SoxS Activator” section on page 7. We have now counted viable cells by plating before and after fermentation, and we observed no decrease in viable cells. We have included this point on page 16 of the manuscript.

19. Page 12 line 1: Have you really identified multiple transcriptional activators? One seems to be more accurate.

We also observed significant activity with TetD and AsiA that could potentially be further optimized. We have adjusted the discussion to clarify this point.

20. Page 12 line 24: What endogenous sites were tested? What modest means? What do you mean by preliminary experiments? Tested once?

We have tested 4 different endogenous genes, using 2-3 gRNA target sites at each gene, with replicate experiments. At most we have observed ~6-fold changes in gene expression by RT-qPCR, which we characterize as “modest” relative to the 50-fold change we observed by RT-qPCR for the GFP reporter (Supplementary Fig. 5). We have several experiments in progress to systematically explore whether it is possible to achieve higher levels of activation. We suggest that these experiments are beyond the scope of the current manuscript.

21. Page 13 line 12: Would this be relatively easy to test? Your best constructs in least another closely-related gram-negative bacterium?

Please see the response to reviewer 1, #10 above. These experiments are in progress in multiple promising bacterial strains. We are currently working to improve our ability to efficiently transform multiple constructs and to tune expression levels in these strains. We view these experiments as beyond the scope of the current manuscript.

22. What is the long-term stability of the constructs? Have you tested over multiple generations? The ethanol experiments seems to indicate that it is relatively stable (I would assume yes if you keep the selective pressure). Have you resequenced the construct after the ethanol fermentation, which appears to have run for 4 days?

To evaluate plasmid stability over the course of the fermentation, we performed minipreps on cells after the fermentation. Analytical restriction digests indicated no large-scale recombination events, and no point mutations were detected by sequencing. We have added this point to the manuscript on page 16.

23. Figure 4: The Y axis values for the GFP expression should all be the same to allow better comparison.

We have made the requested change for all GFP and RFP axes in Fig 4 except the RFP axis in the Ara panel in part C. In general, we observe consistently lower fluorescent protein expression in early stationary phase cultures (Ara inductions) versus late stationary phase cultures (Tet inductions), and it is difficult to see the CRISPRi effect on RFP in this panel if the larger axis scale is used. We have added an explanation to the figure legend to explain this choice.

24. Figure 5: The Y axis values for the GFP expression should all be the same to allow better comparison.

We have made the requested change.

25. References: Bacterial names are not in italic.

We have added italics as appropriate.

26. References: Title of the cited papers will have to be checked for uppercase / lowercase.

We have corrected the uppercase/lowercase issues.

REVIEWERS' COMMENTS:

Reviewer #1 (Remarks to the Author):

Having reviewed the authors' point-by-point responses and the revisions to the manuscript, I believe that authors have satisfactorily and carefully addressed the reviewer comments. I am fully supportive of the revised manuscript.

Reviewer #2 (Remarks to the Author):

none